# Hopf Bifurcation Analysis and Optimal Control of an Infectious Disease with Awareness Campaign and Treatment

**Fahad Al Basir** [1,*] , **Biru Rajak** [2] , **Bootan Rahman** [3] and **Khalid Hattaf** [4]

1   Department of Mathematics, Asansol Girls' College, Asansol-4, West Bengal 713304, India
2   Department of Computer Science, Asansol Girls' College, Asansol-4, West Bengal 713304, India
3   Mathematics Unit, School of Science and Engineering, University of Kurdistan Hewler (UKH), Erbil 44001, Iraq;  bootan.rahman@ukh.edu.krd
4   Equipe de Recherche en Modélisation et Enseignement des Mathématiques (ERMEM), Centre Régional des Métiers de l'Education et de la Formation (CRMEF), Derb Ghalef, Casablanca 20340, Morocco; k.hattaf@yahoo.fr
*   Correspondence: fahadbasir@gmail.com

**Abstract:** Infectious diseases continue to be a significant threat to human health and civilization, and finding effective methods to combat them is crucial. In this paper, we investigate the impact of awareness campaigns and optimal control techniques on infectious diseases without proper vaccines. Specifically, we develop an SIRS-type mathematical model that incorporates awareness campaigns through media and treatment for disease transmission dynamics and control. The model displays two equilibria, a disease-free equilibrium and an endemic equilibrium, and exhibits Hopf bifurcation when the bifurcation parameter exceeds its critical value, causing a switch in the stability of the system. We also propose an optimal control problem that minimizes the cost of control measures while achieving a desired level of disease control. By applying the minimum principle to the optimal control problem, we obtain analytical and numerical results that show how the infection rate of the disease affects the stability of the system and how awareness campaigns and treatment can maintain the stability of the system. This study highlights the importance of awareness campaigns in controlling infectious diseases and demonstrates the effectiveness of optimal control theory in achieving disease control with minimal cost.

**Keywords:** mathematical model; basic reproduction number; stability theory; forward bifurcation; minimum principle; numerical simulations

**MSC:** 49K15; 37L10



## 1. Introduction

Infectious diseases remain a significant public health threat in the modern era despite remarkable advances in science and technology. Microbes such as bacteria, viruses, parasites, and fungi can easily spread from person to person or between humans and animals through various means, including food, water, air, and soil. Deadly infectious diseases include Human Immunodeficiency Virus (HIV), Tuberculosis (TB), and Hepatitis B Virus (HBV). Additionally, diarrhea is a common problem caused by contaminated food or water that can lead to dehydration and death, particularly in low-income countries. According to the World Health Organization, around 17% of deaths worldwide are due to contagious diseases, and HIV/AIDS and TB accounted for 1.5 million deaths in 2019. In the same year, diarrhea was responsible for nearly 300,000 deaths. The recent pandemic caused by SARS CoV-2 has caused around 6.4 million deaths in 2022, making it another deadly infectious disease. Effective prevention and control strategies for infectious diseases are urgently needed to reduce their impact on public health [1–3].

Infectious diseases can have devastating consequences for individuals and communities. To stop the spread of such diseases, basic awareness knowledge is crucial among the population. Health education on different measures can bring the desired change in human behavior, playing a key role in such awareness campaigns [4,5]. An awareness method requires proper technicality and devices to teach the message to the population. The essential elements of such an awareness campaign are prevention by using specific measures, early detection of the problem, and correct treatment under medical supervision. Reaching out to people through different media channels such as social media, mass media, seminars, workshops, and health camps can disseminate the message of disease prevention and control, tailored to the rural or urban setting of the population [6].

Media campaigns can have a significant impact on controlling the spread of infectious diseases. By educating the public about the signs and symptoms of an infection, how it spreads, and what measures can be taken to reduce the risk of transmission, media campaigns can help to reduce the number of new cases and prevent outbreaks from becoming widespread [7,8]. For instance, social media platforms can provide accurate and up-to-date information to the public to dispel myths and avoid the spread of misinformation. Public service announcements (PSAs) on television and radio that educate people about infectious diseases and how to protect themselves and others can also be effective. By educating the public about the importance of handwashing, wearing masks, and maintaining social distancing to prevent the spread of infection, media campaigns can encourage people to take preventive measures [9,10].

In addition, an awareness campaign can promote the importance of vaccination and encourage people to vaccinate in order to reduce their risk of getting sick [11]. Designing and distributing posters and flyers with information about infectious diseases and how to prevent their spread in public places, such as hospitals, schools, shopping malls, and other high-traffic areas, can also be useful [12]. By educating the public about the signs and symptoms of infectious diseases, an awareness campaign can encourage people to seek medical attention if they suspect they may have been exposed. This leads to early detection and treatment, reducing the severity of the illness and preventing its spread to others [13].

It is essential to note that all awareness campaigns should align with the latest guidance from public health authorities and medical professionals to ensure the accuracy and effectiveness of the information being shared. Therefore, media campaigns can be a powerful tool to increase awareness, provide accurate information, and encourage individuals to take preventive measures to stop the spread of infectious diseases.

The impact of media awareness on epidemic outbreaks has been analyzed through model-based mathematical studies [14–19]. These studies examine the disease dynamics of a well-mixed population, where a portion of the susceptible and infected populations are aware of the disease. In these models, aware susceptible individuals are also vulnerable to disease infection but at a lower rate than unaware susceptible individuals.

Another area of study focuses on the impact of information transmission on the dynamics of sexually transmitted infections, assuming that the entire population is aware of the risk, but only a certain proportion is able to respond by limiting their contact with infected individuals [20]. Additionally, epidemic models have been developed that consider the cumulative density of the awareness program as a separate variable [21,22].

This research also takes into account the assumption that infected individuals can recover through awareness-induced treatment and join the aware human population. The model incorporates both local awareness, through information from local people and relatives, and global awareness, through radio and TV campaigns. The level of awareness, $M(t)$, decreases over time as aware individuals become unaware. Aware individuals are assumed to become infected at a lower rate than unaware individuals.

Finally, optimal control theory has been applied to maximize awareness and minimize disease control costs. These studies highlight the important role of media awareness in controlling epidemic outbreaks and provide insights into how awareness campaigns can be optimized to be more effective.

Optimization techniques are a key tool for developing effective control strategies for infectious diseases [23]. One important aspect of this is cost-effectiveness, where a cost function is used to capture the economic and social costs of implementing control measures and the health-related costs of the disease [24,25]. These costs can include things such as the cost of treatment, lost productivity, and the cost of implementing control measures, such as media campaigns [4].

To address this, an optimal control problem is proposed in this paper, which aims to minimize costs by controlling both media campaign costs and treatment costs. The proposed optimization problem is solved using techniques such as Pontryagin's maximum principle, which helps to identify optimal control strategies that balance the costs and benefits of different control measures while taking into account the level of public awareness.

Numerical methods are then used to solve the optimization problem and identify the optimal control strategies. This can help to develop effective and cost-efficient control measures that can be implemented to reduce the spread of infectious diseases. By combining mathematical modeling with optimization techniques, we can develop evidence-based strategies that take into account the complex interactions between the disease, the population, and the various control measures that can be implemented.

The organization of this paper is as follows: Section 2 presents the model and its underlying hypotheses. Section 3 offers the analytical findings, such as the equilibria, stability analysis, and bifurcation analysis. In Section 4, the optimal control problem is formulated. Section 5 presents the numerical simulations and discussions. Finally, in Section 6, this paper concludes by highlighting the benefits and usefulness of the results.

## 2. The Mathematical Model

The mathematical model proposed in this study is based on several assumptions. The model has five variables, where $S(t)$ and $I(t)$ represent the density of susceptible and infected populations at time $t$, respectively.

The interactions between the model variables are shown in Figure 1. The disease is transmitted from infected to susceptible individuals following a mass action functional form. A media campaign increases 'level of awareness' denoted as $M(t)$. Campaign can be carried out to increase awareness in the unaware susceptible population. This campaign divides the total susceptible population into two subclasses: the unaware susceptible population $S_u$ and the aware susceptible population $S_a$. As awareness disseminates, people change their behavior to alter their susceptibility. It is also assumed that infected individuals recover through treatment at a rate $r$, and after recovery, a fraction $p$ of recovered people join the unaware susceptible class while the remaining fraction $q = (1 - p)$ join the aware susceptible class at a rate $\gamma$.

Several parameters are used to describe the model. These include $b$, which is the constant recruitment rate in the susceptible population; $\lambda$, the disease transmission rate; $d$, the natural mortality rate of the population; $\delta$, the disease-induced mortality rate of the infected population; and $r$, the recovery rate. The disease spreads due to direct contact between susceptible and infective individuals at a rate $\lambda$, and the transfer rate from aware class to unaware class is denoted as $\beta$.

All newly recruited individuals are assumed to be unaware, and the rate of being aware is proportional to the number of infected individuals reported by the media and/or health organization. The depletion of the aware state is inversely proportional to the number of cases. Unaware susceptible individuals become aware susceptible at the rate of $\alpha M$, where $\alpha$ is the maximum rate at which an unaware susceptible individual becomes aware susceptible. On the other hand, aware susceptible individuals become unaware susceptible at a rate of $\beta/(1 + M)$ due to memory fading and/or carelessness. Level of awareness increases when awareness programs are implemented proportionally with the change in unaware infective individuals at a rate of $\eta$ and cut down at a rate of $\theta$ due to

their ineffectiveness. Level of awareness also increases from the awareness campaign by global sources, such as radio, TV, etc. [21], at a constant rate $\omega$.

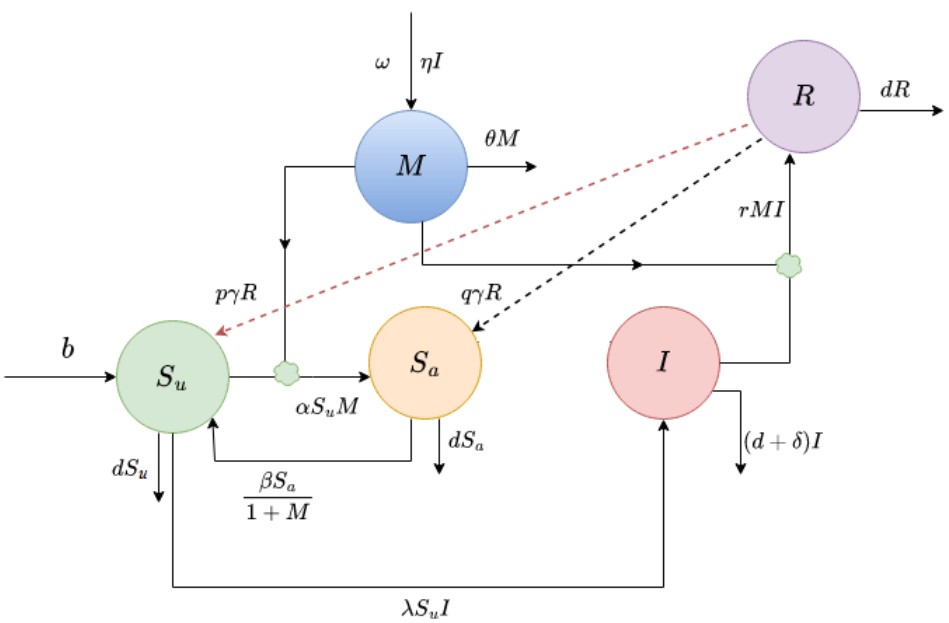

**Figure 1.** Interactions between populations is shown.

The above assumptions lead to the following model:

$$
\begin{aligned}
\frac{dS_u}{dt} &= b - \alpha M S_u - \lambda S_u I + \frac{\beta S_a}{1+M} - dS_u + p\gamma R, \\
\frac{dS_a}{dt} &= \alpha M S_u - \frac{\beta S_a}{1+M} - dS_a + q\gamma R, \\
\frac{dI}{dt} &= \lambda S_u I - (d+\delta)I - rMI, \\
\frac{dR}{dt} &= rMI - dR - \gamma R, \\
\frac{dM}{dt} &= \omega + \eta I - \theta M,
\end{aligned}
\tag{1}
$$

with the initial conditions:

$$
S_u(0) > 0, S_a(0) > 0, I(0) > 0, R(0) > 0, M(0) > 0.
\tag{2}
$$

For the analysis of model (1), the region of attraction is given by the set

$$
\mathcal{B} = \left\{ (S_u, S_a, I, R, M) \in R_+^5 : 0 \le S_u + S_a + I + R \le \frac{b}{d}, 0 \le M \le \frac{\omega d + \eta b}{d\theta} \right\}.
\tag{3}
$$

Thus, all solutions of the model (1) are bounded in $\mathcal{B}$ for all $t > 0$.

### 2.1. Basic Properties of the Model

In this subsection, the basic properties of the system (1) such as existence and non-negativity of solutions are discussed.

Because the right-hand side of (1) is locally Lipschitz, we deduce from the standard theory of functional differential equations [26] that system (1) admits a unique solution.

The $S_u, S_a, I, R, M$ for the effect of awareness on the transmission dynamics of infectious disease will be analyzed in a biologically and mathematically viable region as

follows. This region should be feasible for all population. For this, the following theorem is established.

**Theorem 1.** *All solutions of system* (1) *with initial conditions in* (2) *are positive for all* $t > 0$.

**Proof.** Let

$$T_1 = \sup\{t > 0 : H_u(t) > 0, H_a(t) > 0, H_i(t) > 0, V_s(t) > 0, V_i(t) > 0, M(t) > 0\}.$$

Because $H_u(0) > 0, H_a(0) > 0, H_i(0) > 0, V_s(0) > 0, V_i(0) > 0$, and $M(0) > 0$, then $T_1 > 0$. If $T_1 < \infty$, then $H_u, H_a, H_i, V_s, V_i, M$ are all equal to zero at $T_1$.

It follows from the first equation of the system (1) that

$$\frac{dS_u}{dt} = b - \alpha M S_u - \lambda S_u I + \frac{\beta S_a}{1 + M} - dS_u + p\gamma R.$$

That is,

$$\frac{dS_u}{dt} + (\alpha M + \lambda I + d)S_u = \left(b + \frac{\beta S_a}{1 + M} + p\gamma R\right)$$

Thus,

$$\frac{d}{dt}\left\{S_u(t)\exp\left[\int_0^t (\alpha M(\xi) + \lambda I(\xi) + d)d\xi\right]\right\}$$
$$= \left(b + \frac{\beta S_a}{1 + M} + p\gamma R\right)\exp\left[\int_0^t (\alpha M(\xi) + \lambda I(\xi) + d)d\xi\right].$$

Hence,

$$S_u(T_1)\exp\left[\int_0^t (\alpha M(\xi) + \lambda I(\xi) + d)d\xi\right] - S_u(0)$$
$$= \int_0^{T_1}\left\{\left(b + \frac{\beta S_a}{1 + M} + p\gamma R\right)\exp\left[\int_0^t (\alpha M(\xi) + \lambda I(\xi) + d)d\xi\right]\right\}dv$$

So that,

$$S_u(T_1) = S_u(0)\exp\left[\int_0^t (\alpha M(\xi) + \lambda I(\xi) + d)d\xi\right]$$
$$+ \exp\left[\int_0^t (\alpha M(\xi) + \lambda I(\xi) + d)d\xi\right]$$
$$\times \int_0^{T_1}\left\{\left(b + \frac{\beta S_a}{1 + M} + p\gamma R\right)\exp\left[\int_0^t [\alpha M(\xi) + \lambda I(\xi) + d]d\xi\right]\right\}dv > 0$$

Following the same procedure, it can be shown that the rest of the model populations are positive for all $t > 0$. Thus, $\mathbb{R}_+^5 = \{(S_u, S_a, I, R, M)|S_u \geq 0, S_a \geq 0, I \geq 0, R \geq 0\}$ is an invariant under the flow of the system (1). □

### 2.2. The Basic Reproduction Number

We follow the method established in the paper by Heffernan et al. [27] for calculating $R_0$. We consider the next generation matrix G which comprises two parts, namely, $F$ and $V$, where

$$F = \left[\frac{\partial F_i(E_0)}{\partial x_j}\right] = \begin{bmatrix} \lambda \bar{S}_u & 0 \\ r\bar{M} & 0 \end{bmatrix}$$

$$V = \left[\frac{\partial V_i(E_0)}{\partial x_j}\right] = \begin{bmatrix} (d + \delta) + r\bar{M} & 0 \\ 0 & \gamma + d \end{bmatrix}$$



where $F_i$ are the new infections, while $V_i$ transfers infections from one compartment to another. $E_0$ is the disease-free equilibrium. We obtain $R_0 = \frac{\lambda \bar{S}_u}{(d+\delta)+r\bar{M}}$ and it is the dominant eigenvalue of the matrix $G = FV^{-1}$.

## 3. Dynamics of the System

The system (1) is a set of nonlinear ordinary differential equations. In order to understand the dynamics of the system, we can analyze its steady states, which are the values of the variables at which they do not change with time.

### 3.1. Existence of Equilibria

The system has two equilibria. Namely, the disease-free steady state $E_0$ is always existing and is given by $E^0(\bar{S}_u, \bar{S}_a, 0, 0, \bar{M})$, where

$$
\begin{aligned}
\bar{M} &= \frac{\omega}{\theta}, \\
\bar{S}_u &= \frac{b\theta(d\omega + \beta\theta + d\theta)}{d(\alpha\omega^2 + \alpha\omega\theta + d\omega\theta + \beta\theta^2 + d\theta^2)}, \\
S_a &= \frac{\alpha b\omega(\omega + \theta)}{d(\alpha\omega^2 + \alpha\omega\theta + d\omega\theta + \beta\theta^2 + d\theta^2)},
\end{aligned}
\tag{4}
$$

and the endemic equilibrium point $E^*(S_u^*, S_a^*, I^*, R^*, M^*)$ where

$$
\begin{aligned}
I^* &= \frac{(\theta M^* - \omega)}{\eta}, \quad R^* = \frac{rM^*I^*}{d+\delta}, \quad S_u^* = \frac{rM^* + (d+\delta)}{\lambda}, \\
S_a^* &= \frac{(\alpha S_u^* M^* + q\gamma R^*)(1+M^*)}{\beta + d(1+M^*)},
\end{aligned}
$$

and $M^*$ is the positive root of the following equation:

$$
c_0 M^3 + c_1 M^2 + c_2 M + c_3 = 0,
\tag{5}
$$

where

$$
\begin{aligned}
c_0 =\ & -\alpha d^2\eta r - d^2\lambda r\theta - \alpha d\eta\gamma r - d\gamma\lambda qr\theta, \\
c_1 =\ & -\alpha d^3\eta - d^3\eta r - d^3\lambda\theta - \alpha d^2\delta\eta - \alpha d^2\eta\gamma - \alpha d^2\eta r - d^2\eta\gamma r - d^2\delta\lambda\theta - d^2\gamma\lambda\theta + d^2\lambda\omega r \\
& -d^2\lambda r\theta - \alpha d\delta\eta\gamma - \alpha d\eta\gamma r - d\delta\gamma\lambda\theta - b\eta d\lambda r\theta + d\gamma\lambda\omega qr - d\gamma\lambda qr\theta, \\
c_2 =\ & -d^4\eta - \alpha d^3\eta + d^2\gamma\lambda\omega - d^3\delta\eta - d^3\eta\gamma - d^3\eta r + d^3\lambda\omega - d^3\lambda\theta - \alpha d^2\delta\eta \\
& -\alpha d^2\eta\gamma - d^2\delta\eta\gamma + bd^2\eta\lambda - b\eta d^2\eta r - d^2\eta\gamma r + d^2\delta\lambda\omega - b\eta d^2\lambda\theta + d^2\gamma\lambda\omega \\
& -d^2\delta\lambda\theta - d^2\gamma\lambda\theta + d^2\lambda\omega r - \alpha d\delta\eta\gamma + bd\eta\gamma\lambda - b\eta d\eta\gamma r - b\eta d\delta\lambda\theta + d\delta\gamma\lambda\omega \\
& -b\eta d\gamma\lambda\theta - b\eta\delta\gamma\lambda\theta - d\delta\gamma\lambda\theta + b\eta d\lambda\omega r + d\gamma\lambda\omega qr, \\
c_3 =\ & -d^4\eta - b\eta d^3\eta - d^3\delta\eta - d^3\eta\gamma + d^3\lambda\omega - b\eta d^2\delta\eta - b\eta d^2\eta\gamma - d^2\delta\eta\gamma + bd^2\eta\lambda \\
& +b\eta d^2\lambda\omega + d^2\delta\lambda\omega - b\eta d\delta\eta\gamma + bb\eta d\eta\lambda + bb\eta\eta\gamma\lambda + bd\eta\gamma\lambda + b\eta d\delta\lambda\omega \\
& +b\eta d\gamma\lambda\omega + b\eta\delta\gamma\lambda\omega + d\delta\gamma\lambda\omega.
\end{aligned}
$$

It should be noted that because $c_0 < 0$, the following cases can be distinguished: (i) if $c_2 > 0$ and $c_3 > 0$, there exists a unique endemic equilibrium; (ii) if $c_1 < 0$, $c_2 > 0$, and $c_3 < 0$, then (5) has two positive roots, resulting in the possibility of two feasible endemic equilibria; and (iii) if $c_1 > 0$, $c_2 < 0$, and $c_3 > 0$, then (5) has three roots, leading to the possibility of three feasible endemic equilibria. $I^*$ is positive when $\theta M^* > \frac{\omega}{\theta}$.

*3.2. Stability Analysis of $E_0$*

**Theorem 2.** *For the system (1), the disease-free steady state $E_0$ is stable if $R_0 < 1$ and unstable if $R_0 > 1$.*

**Proof.** The Jacobian matrix at the disease-free equilibrium $E_0$ is

$$A(E_0) = \begin{bmatrix} -\alpha\bar{M} - d & \frac{\beta}{1+\bar{M}} & -\lambda\bar{S}_u & p\gamma & -\frac{\beta}{1+\bar{M}} \\ \alpha\bar{M} & -\frac{\beta}{1+\bar{M}} - d & 0 & q\gamma & \frac{\beta}{1+\bar{M}} \\ 0 & 0 & \lambda\bar{S}_u - r\bar{M} - (d+\delta) & 0 & 0 \\ 0 & 0 & r\bar{M} & -d & 0 \\ 0 & 0 & \eta & 0 & -\theta \end{bmatrix},$$

At the disease-free steady state $E_0$, the eigenvalues are $\rho_1 = -\theta$, $\rho_2 = -d$ which are both negative. Additionally, $\rho_3 = \lambda\bar{S}_u - r\bar{M} - (d+\delta)$ is also negative when the basic reproduction number $R_0 < 1$. The remaining eigenvalues can be found by solving the given equation:

$$\rho^2 + \left(\alpha\bar{M} + 2d + \frac{\beta}{1+\bar{M}}\right)\rho + \left[(\bar{M}+d)\left(d + \frac{\beta}{1+\bar{M}} + d\right)\right] = 0.$$

The expression for $\bar{M}$ is provided in (4). Because the coefficients of (6) are positive, the roots of (6) denoted by $\rho_4$ and $\rho_5$ are therefore negative or have negative real parts. Therefore, the stability of $E_0$ is guaranteed only when $\rho_3 < 0$. □

**Remark 1.** *The detailed numerical calculations indicate that the interior equilibrium $E^*$ is feasible when $R_0 > 1$, i.e., when the disease-free equilibrium $E_0$ is unstable. Moreover, at $R_0 = 1$, the system undergoes a forward transcritical bifurcation.*

Therefore, based on the definition of $R_0$ and the stability analysis of $E_0$, we can state the following theorem.

*3.3. Stability of $E^*$ and Hopf Bifurcation*

The Jacobian matrix $J_{5\times5}$ at the endemic equilibrium point $E^*(S_u^*, S_a^*, I^*, R^*, M^*)$ is determined for the stability analysis of this equilibrium point and is given below by

$$J(E^*) = [J_{ij}]_{5\times5} = \begin{bmatrix} -\alpha M^* - \lambda I^* - d & \frac{\beta}{1+M^*} & -\lambda S_u^* & p\gamma & -\frac{\beta}{(1+M^*)^2} \\ \alpha M^* & -\frac{\beta}{1+M^*} - d & 0 & q\gamma & \frac{\beta}{(1+M^*)^2} \\ \lambda I^* & 0 & \lambda S_u^* - r M^* - (d+\delta) & 0 & -r I^* \\ 0 & 0 & r M^* & -d-\gamma & r I^* \\ 0 & 0 & \eta & 0 & -\theta \end{bmatrix},$$

The characteristic equation is given by

$$x^5 + a_1 x^4 + a_2 x^3 + a_3 x^2 + a_4 x + a_5 = 0, \tag{6}$$

where

$$
\begin{aligned}
a_1 =\ & -(J_{11} + J_{22} + J_{33} + J_{44} + J_{55}), \\
a_2 =\ & -J_{12}J_{21} - J_{13}J_{31} + J_{11}(J_{22} + J_{33} + J_{44}) + J_{22}(J_{33} + J_{44}) \\
& + J_{33}J_{44} - J_{35}J_{53} + J_{55}(J_{11} + J_{22} + J_{33} + J_{44}), \\
a_3 =\ & J_{31}(J_{13}J_{22} - J_{12}J_{23} - J_{14}J_{43}) + J_{33}(J_{12}J_{21} - J_{11}J_{22}) + J_{44}(J_{12}J_{21} - \\
& J_{11}J_{22} + J_{13}J_{31} - J_{11}J_{33} - J_{22}J_{33}) + J_{53}(-J_{15}J_{31} + J_{11}J_{35} + \\
& J_{22}J_{35} + J_{35}J_{44})J_{53} + J_{55}(J_{12}J_{21} + -J_{11}J_{22} + J_{13}J_{31}) - J_{33}J_{44}(J_{11} + J_{22}) \\
& - J_{44}J_{55}(J_{11} + J_{22} + J_{33}), \\
a_4 =\ & J_{31}J_{43}(J_{14}J_{22} - J_{12}J_{24}) + J_{31}J_{44}(J_{12}J_{23} - J_{13}J_{22}) + J_{33}J_{44}(J_{11}J_{22} - J_{12}J_{21}) + \\
& J_{31}J_{53}(J_{15}J_{22} - J_{12}J_{25}) + J_{35}J_{53}(J_{12}J_{21} - J_{11}J_{22}) + J_{44}J_{53}(J_{15}J_{31} - J_{11}J_{35}) \\
& - J_{31}J_{55}(J_{22}J_{35}J_{44} + J_{14}J_{31}J_{45})J_{53} + J_{31}J_{55}(J_{12}J_{23} - J_{13}J_{22}) + J_{33}J_{55}(J_{11}J_{22} - J_{12}J_{21}) + \\
& J_{14}J_{31}J_{43}J_{55} + J_{44}J_{55}(-J_{12}J_{21} + J_{11}J_{22} - J_{13}J_{31}) + J_{33}J_{44}J_{55}(J_{11} + J_{22}), \\
a_5 =\ & J_{31}J_{44}J_{53}(J_{15}J_{22} + J_{12}J_{25}) + (-J_{12}J_{21} + J_{11}J_{22})J_{35}J_{44}J_{53} \\
& + J_{31}J_{45}J_{53}(J_{14}J_{22} - J_{12}J_{24}) + J_{31}J_{43}J_{55}(-J_{14}J_{22} + J_{12}J_{24}) \\
& + J_{31}J_{44}J_{55}(J_{13}J_{22} - J_{12}J_{23}) \\
& + (J_{12}J_{21} - J_{11}J_{22})J_{33}J_{44}J_{55}.
\end{aligned}
$$

If the following conditions hold, all roots of the characteristic equation have a negative real part according to the Routh–Hurwitz criterion:

$$
a_5 > 0,\ a_1 a_2 - a_3 > 0,\ a_3(a_1 a_2 - a_3) - a_1(a_1 a_4 - a_5) > 0
$$
$$
\text{and}\ (a_1 a_2 - a_3)(a_3 a_4 - a_2 a_5) + (a_1 a_4 - a_5)(a_5 - a_1 a_4) > 0. \tag{7}
$$

Thus, endemic equilibrium $E^*$ is stable if the conditions in (7) hold. The following theorem ensures the occurrence of Hopf bifurcation at the endemic equilibrium.

**Theorem 3.** *The stability of the interior equilibrium $E^*$ depends on whether the conditions in (7) are satisfied. If the conditions are met, then $E^*$ is stable; otherwise, it becomes unstable. Moreover, $E^*$ undergoes a Hopf bifurcation at the critical value $\theta^*$ of the generic parameter $\theta$ if either of the following conditions are satisfied:*

i.　$\phi(\theta^*) = 0$ *and* $\left.\dfrac{d\phi}{d\theta}\right|_{\theta=\theta^*} \neq 0$, *where*

$$
\phi(\theta) = (a_3 - a_1 a_2)(a_5 a_2 - a_3 a_4) - (a_5 - a_1 a_4)^2,
$$

　　*with*

$$
\varphi = \frac{a_5 - a_1 a_4}{a_3 - a_1 a_2} > 0, \quad a_3 - a_1 \varphi \neq 0,
$$

ii.　$a_5 = a_1 a_4$, $a_3 = a_1 a_2$, $a_4 < 0$, $a_1 a_3 \neq 0$,

$$
\left[a_1' \varphi^2 + (a_1 a_2' - a_3')\varphi - (a_1 a_4' - a_5')\right]\Big|_{\theta=\theta^*} \neq 0.
$$

　　*where $a_j'(j = 1, 2, \ldots, 5)$ represent the derivatives of $a_j(j = 1, 2, \ldots, 5)$ with respect to the genic bifurcation parameter $\theta$ and*

$$
\varphi = \frac{1}{2}\left(a_2 + \sqrt{a_2^2 - 4a_4}\right) > 0.
$$

**Proof.** We need the following lemma (lemma 2 of [28,29]) for proofing Theorem 3.

**Lemma 1.** *Suppose that the following conditions are satisfied:*

$$(A_3 - A_1 A_2)(A_2 A_5 - A_3 A_4) - (A_5 - A_1 A_4)^2 > 0. \tag{8}$$

*and*

$$A_1 > 0, A_1 A_2 - A_3 > 0, A_3 - A_1 \theta > 0. \tag{9}$$

*Then, the polynomial $H(\xi)$ possesses at least one pair of purely imaginary roots $\xi_{1,2} = \pm i\sqrt{\theta}, \theta > 0$ and the rest of the roots with negative real parts.*

We now provide the proof of Theorem 3 below.

The first two conditions of the theorem are fulfilled from Lemma 1. We have to prove the last condition only, which is the transversality condition.

Using the well-known Vieta formulas, given the coefficients of the following polynomial,

$$H(\xi) = \xi^5 + A_1\xi^4 + A_2\xi^3 + A_3\xi^2 + A_4\xi + A_5 \tag{10}$$

from its roots, it follows that the function $\Psi(\zeta)$ of conditions (i) can be written in the form of Orlando's formula:

$$\begin{aligned} \Psi(\zeta) &= (\xi_1 + \xi_2)(\xi_1 + \xi_3)(\xi_1 + \xi_4)(\xi_1 + \xi_5)(\xi_2 + \xi_3)(\xi_2 + \xi_4)(\xi_2 + \xi_5) \times \\ &\quad (\xi_3 + \xi_4)(\xi_3 + \xi_5)(\xi_4 + \xi_5). \end{aligned} \tag{11}$$

Assume that two roots $\xi_1$ and $\xi_2$ of (10) are in the form

$$\xi_{1,2} = \chi(\zeta) \pm i\nu(\zeta). \tag{12}$$

with $\chi(\zeta^*) = 0$ and $\nu(\zeta^*) = \sqrt{\theta}$, while $Re(\xi_{3,4,5}) \neq 0$. Then, we have

$$\begin{aligned} \Phi(\zeta) &= 2\chi[(\chi + \xi_3)^2 + \nu][(\chi + \xi_4)^2 + \nu](\chi + \xi_5)^2 + \nu] \times \\ &\quad (\xi_3 + \xi_4)(\xi_3 + \xi_5)(\xi_4 + \xi_5). \end{aligned} \tag{13}$$

with $\Phi(\zeta^*) = 0$.

Differentiating $\Phi(\zeta)$ with respect to $\zeta$ and putting $\zeta = \zeta^*$, we finally obtain

$$\begin{aligned} \frac{d\Phi(\zeta)}{d\zeta}\bigg|_{\zeta=\zeta^*} &= \bigg\{ 2(\nu^2 + \xi_3^2)(\nu^2 + \xi_4^2)(\nu^2 + \xi_5^2) \times \\ &\quad (\xi_3 + \xi_4)(\xi_3 + \xi_5)(\xi_4 + \xi_5)\frac{d\chi(\zeta)}{d\zeta} \bigg\}\bigg|_{\zeta=\zeta^*} \end{aligned} \tag{14}$$

If the condition (9) is true, then the roots $\xi_3, \xi_4, \xi_5$ have negative real parts at $\zeta = \zeta^*$ and only the last factor in (14) may possibly be zero.

Thus,

$$\frac{d\chi(\zeta)}{d\zeta}\bigg|_{\zeta=\zeta^*} \neq 0 \Leftrightarrow \frac{d\Phi(\zeta)}{d\zeta}\bigg|_{\zeta=\zeta^*} \tag{15}$$

Therefore, the transversality condition is satisfied. Thus, Hopf bifurcation occurs at $\zeta = \zeta^*$. □

## 4. The Optimal Control Problem

In this section, the aim is to study the impact of an optimal treatment and awareness campaign. For this, there are two control parameters, $u_1(t)$ (for treatment cost) and $u_2(t)$

(for the cost of awareness campaign). Based on the above assumptions, the above system (1) would be

$$
\begin{aligned}
\frac{dS_u}{dt} &= b - \alpha M S_u - \lambda S_u I + \frac{\beta S_a}{1 + M} - d S_u + p\gamma R, \\
\frac{dS_a}{dt} &= \alpha M S_u - \frac{\beta S_a}{1 + M} - d S_a + q\gamma R \\
\frac{dI}{dt} &= \lambda S_u I - (d + \delta)I - u_1(t)rMI \\
\frac{dR}{dt} &= u_1(t)rMI - dR - \gamma R, \\
\frac{dM}{dt} &= (1 - u_2(t))\omega + \eta I - \theta M,
\end{aligned}
\tag{16}
$$

For later use, we rewrite the system (16) in the following form:

$$
\frac{dx_i}{dt} = f_i(t, x),
\tag{17}
$$

where $f_i$ ($i = 1, \ldots, 5$) are the right sides of system (16) and $x_j$ ($j = 1, \ldots, 5$) are the state variables corresponding to $S_u, S_a, I, R, M$, respectively.

Our objective is to minimize the cost of the media campaign, treatment, and number of infected cases. Therefore, we formulate the cost function as follows:

$$
J[u_1(t), u_2(t)] = \int_0^{t_f} [A u_1^2(t) + B u_2^2(t) - P M^2 + Q I^2(t)] dt.
\tag{18}
$$

$t_f$ is the final time. The parameters $P > 0$), $Q > 0$ are the weight constants on the benefit of the costs and $A > 0$, $B > 0$ are the penalty multipliers.

Now, the objective is to find the optimal control pair $u^*(t) = (u_1(t), u_2(t))$ such that

$$
\begin{aligned}
J(u_1^*, u_2^*) = \quad & min\,(J(u_1, u_2) : (u_1, u_2) \in U) \text{ where } U = U_1 \times U_2, \\
& \text{where } U_1 = (u_1(t) : u_1 \text{ is measurable and } 0 \le u_1 \le 1,\ t \in [0, t_f]) \text{ and} \\
& U_2 = (u_2(t) : u_2 \text{ is measurable and } 0 \le u_2 \le 1,\ t \in [0, t_f]).
\end{aligned}
$$

Here, Pontryagin minimum principle [30] has been used to find the optimal control pair $(u_1^*(t),\ u_2^*(t))$.

The necessary conditions for the optimal control problem can be obtained using Pontryagin's minimum principle, as presented in [30]. Applying this principle to the system, we obtain the following theorem.

**Theorem 4.** *Suppose the given optimal control pair $u^*(t) = (u_1^*(t), u_2^*(t))$ and the solution $(S_u^*(t), S_a^*(t), I^*(t), R^*(t), M^*(t))$ of the corresponding system (16) minimize $J(u^*)$ over U. Then, by applying Pontryagin's minimum principle in state, the following theorem holds: there exist adjoint variables $\xi_1, \xi_2, \xi_3, \xi_4,$ and $\xi_5$ that satisfy the following equations:*

$$
\begin{aligned}
\frac{d\xi_1}{dt} &= -(\xi_1 J_{11} + \xi_2 J_{21} + \xi_3 J_{31}), \\
\frac{d\xi_2}{dt} &= -(\xi_1 J_{12} + \xi_2 J_{22}), \\
\frac{d\xi_3}{dt} &= -2PI - (\xi_1 J_{13} + \xi_2 J_{23} + \xi_3 J_{33} + \xi_4 J_{43} + \xi_5 J_{53}), \\
\frac{d\xi_4}{dt} &= -(\xi_1 J_{14} + \xi_2 J_{24} + \xi_3 J_{34} + \xi_4 J_{44} + \xi_5 J_{54}), \\
\frac{d\xi_5}{dt} &= 2QM - (\xi_1 J_{15} + \xi_2 J_{25} + \xi_3 J_{35} + \xi_4 J_{45} + \xi_5 J_{55}).
\end{aligned}
\tag{19}
$$

*along with the boundary conditions, $\xi_i(t_f) = 0$, $i = 1, \ldots, 5$.*

**Proof.** The Hamiltonian for the optimal control problem can be taken as

$$H = Au_1^2(t) + B(1 - u_2(t))^2 - PM^2(t) + QI^2 + \sum_{i=1}^{6} \xi_i f_i, \ i = 1, 2, \ldots, 6. \tag{20}$$

According to the maximum principle [30], the unconstrained optimal control pair $u^* = (u_1^*(t), u_2^*(t))$ satisfies

$$\frac{\partial H}{\partial u^*} = 0, \ u^* = (u_1^*, u_2^*). \tag{21}$$

Thus, from (21), we have

$$u_1^*(t) = \frac{(\xi_i 3 - \xi_4)rMI}{2A}, \tag{22}$$

$$u_2^*(t) = \frac{2B + \xi_5 \omega}{2B}. \tag{23}$$

Due to the boundedness of the optimal control parameters, we have the following forms of the optimal control pair:

$$u_1^*(t) = max(0, \ min(1, \ \frac{(\xi_i 3 - \xi_4)rMI}{2A})), \tag{24}$$

$$u_2^*(t) = max(0, \ min(1, \ \frac{2B + \xi_5 \omega}{2B})). \tag{25}$$

According to minimum principle [30], we have the following relation for determining the adjoint system:

$$\frac{d\xi_i}{dt} = -\frac{\partial H}{\partial x_i}, \ i = 1, 2, 3, 4, 5, \tag{26}$$

where $x_i \equiv (S_u, S_a, I, R, M)$ and the necessary condition satisfying the optimal control pair $u^*(t)$ is

$$H(x_i(t), u^*(t), \xi_i(t), t) = \min_{u \in U}(H(x_i(t), u(t), \xi_i(t), t)), i = 1, 2, 3, 4, 5, 6. \tag{27}$$

So, the adjoint system (19) corresponding to the system (16) can be obtained by Equation (26). The boundary conditions for the adjoint system (19) are $\xi_i(t_f) = 0$, $(i = 1, \ldots, 6)$ as the salvage function in the objective functional (18) is assumed to be zero. □

## 5. Numerical Simulation

This section presents the numerical simulations conducted to investigate the system dynamics. The goal is to examine the impact of increasing the infection rate on the system's behavior using simulations with and without control, in order to verify the theoretical findings.

Forward bifurcation is shown in Figure 2. This figure shows that the disease-free equilibrium $E_0$ is stable for $R_0 < 1$ and unstable for $R_0$. Consequently, a transcritical forward bifurcation occurs at $R_0 = 1$. The region of stability of $E_0$ is shown in the $\alpha - \omega$ and $\lambda - \omega$ parameter planes in Figure 3a,b respectively. When the awareness rates are high or the infection rate is low, $E_0$ is stable as the value of $R_0$ is below unity. On the boundary of the stable and unstable region, the value of $R_0$ is unity. That means the forward bifurcation points lie on this boundary.

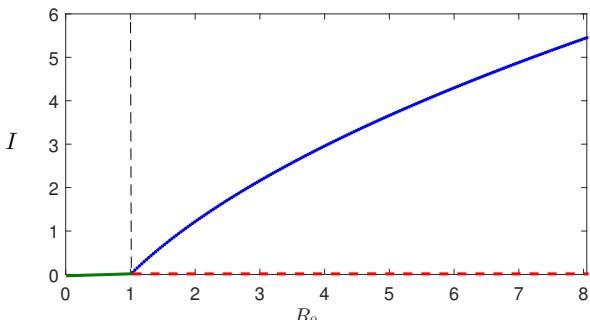

**Figure 2.** Forward bifurcation is shown. $\lambda$ is varied in the interval $(0, 0.0005)$ and the rest of the parameter values are taken from Table 1.

**Table 1.** List of parameters used for numerical simulations.

| Parameter | Definition | Reference | Value (day$^{-1}$) |
|---|---|---|---|
| $b$ | Constant recruitment rate | [12] | 12 |
| $\lambda$ | Disease transmission rate | [22,31] | 0.0005 |
| $\alpha$ | Contact rate between unaware susceptible with media | [12] | 0.002 |
| $\omega$ | Rate of media campaigns by global sources | [21,22] | 0.03 |
| $d$ | Susceptible class natural death rate | [12,32] | 0.01 |
| $\delta$ | Additional death rate due to infection | [32] | 0.007 |
| $\beta$ | Rate at which aware human becomes unaware | [22] | 0.0025 |
| $r$ | Rate of recovery of infected human | [12] | 0.01 |
| $\gamma$ | Rate at which recovered class becomes susceptible after immunity loss | [12] | 0.0015 |
| $p$ | Portion of recovered class becoming susceptible unaware class | [12] | 0.3 |
| $\eta$ | Rate of awareness programs by local sources | [12] | 0.25 |
| $\theta$ | Depletion rate of awareness program | [12,22] | 0.015 |

Figure 4a–e show the solution trajectories for two different values of $\lambda$. For lower values of $\lambda$, the model populations oscillate initially and then converge to the endemic equilibrium $E^*$. However, when $\lambda$ exceeds a certain threshold value $\lambda^* = 0.000495$, all populations exhibit periodic oscillations, indicating that they bifurcate into periodic solutions. Figure 4f shows a stable (supercritical) limit cycle is observed for $\lambda = 0.0005$. The bifurcation diagram for the maximum and minimum values of the periodic solutions is shown in Figure 5, where we observe that the stability switch occurs at $\lambda = \lambda^*$.

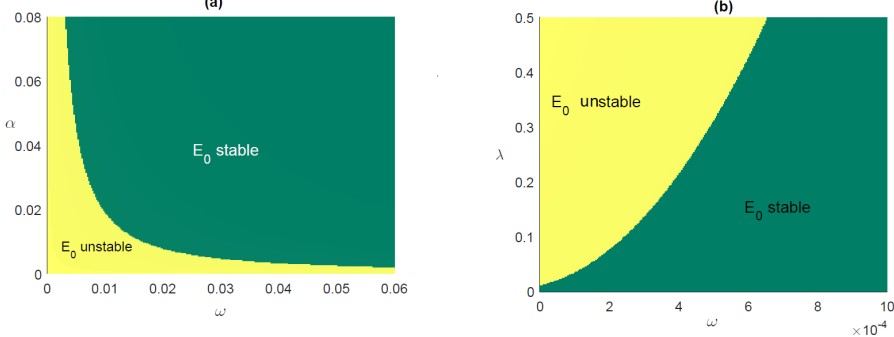

**Figure 3.** Stability of $E_0$ in (**a**) $\alpha - \omega$, (**b**) $\lambda - \omega$ parameter planes. Other parameter values are taken from Table 1.

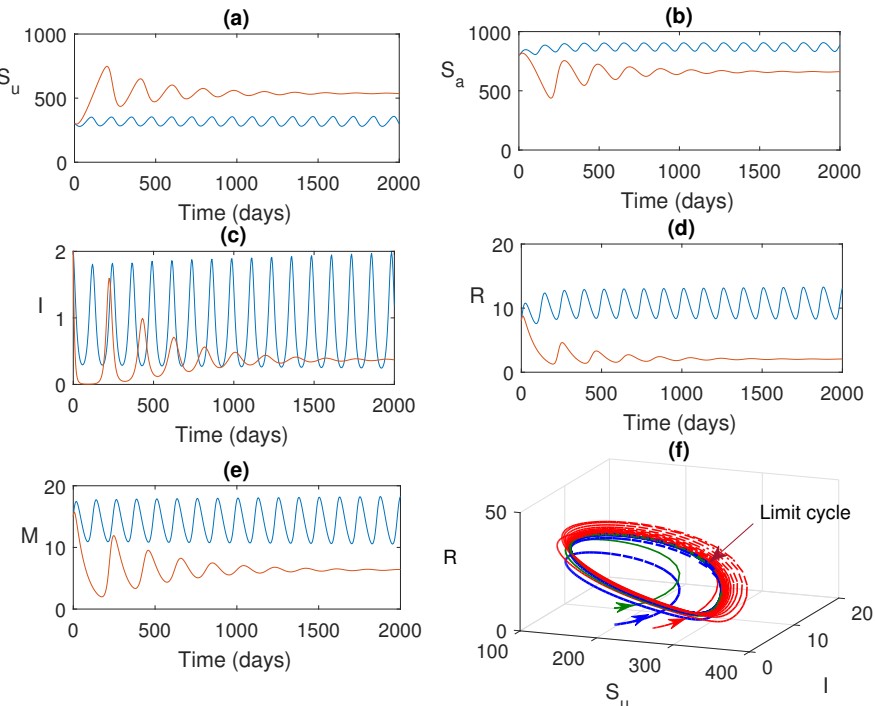

**Figure 4.** (**a**–**e**): Time series solution of the system (1) is plotted for $\lambda = 0.0001$ (red line) and $\lambda = 0.0005$ (blue line). Parameter values are as given in Table 1. (**f**): Limit cycle is shown in $S_u - I - R$ plane.

We also explore the bifurcation of $\omega$ in Figure 6 for a fixed value of $\lambda = 0.001$ for which periodic oscillations exist. As we increase the value of $\omega$, the unstable endemic state becomes stable when $\omega$ exceeds a threshold value $\omega^* = 0.149$.

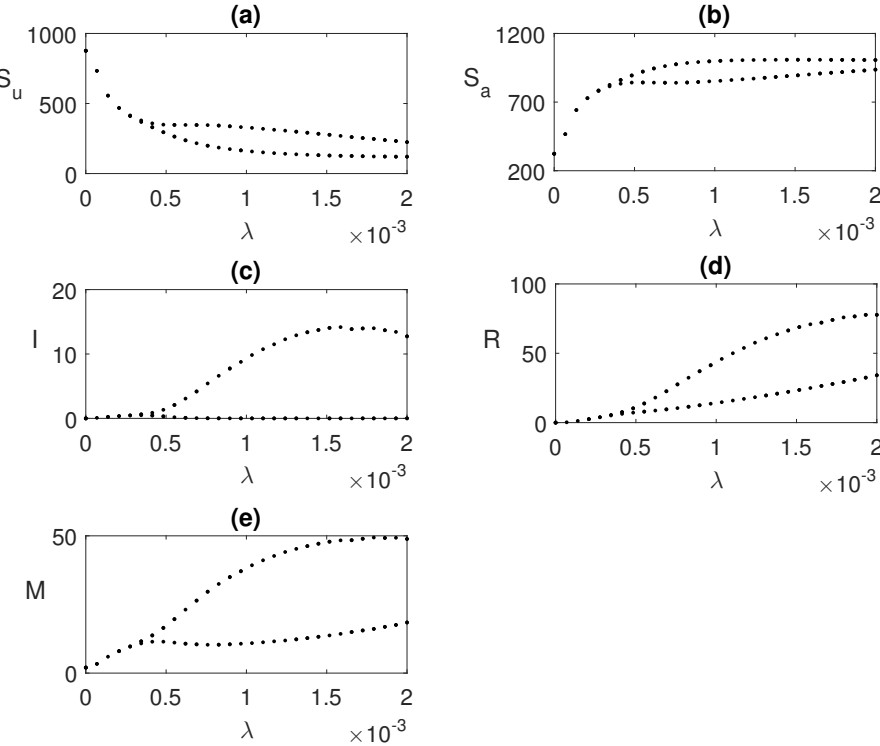

**Figure 5.** (**a**–**e**): Hopf bifurcation taking $\lambda$ as main parameter. Values of the parameters are same as Figure 4.

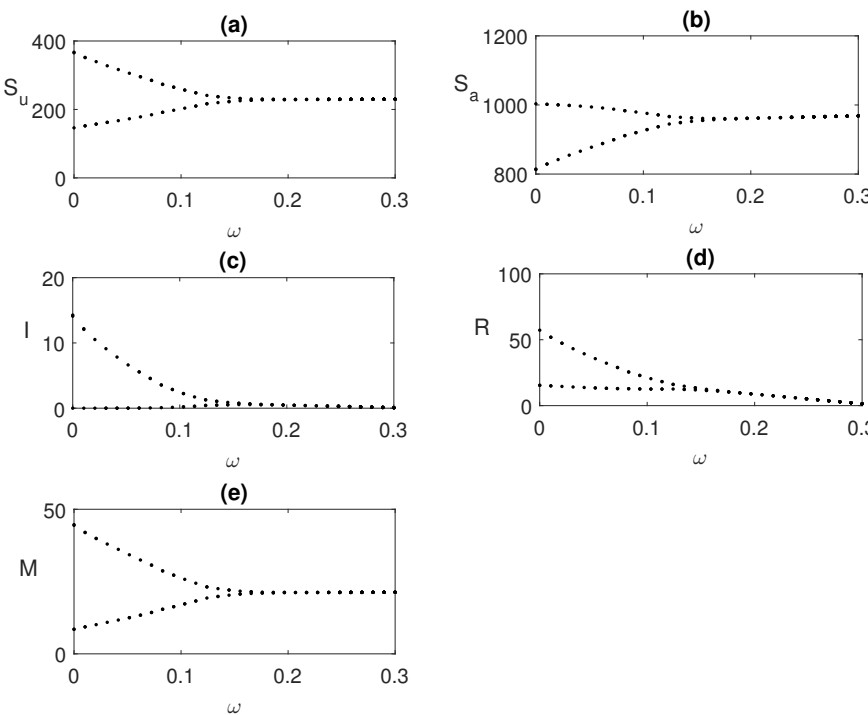

**Figure 6.** (**a**–**e**): Hopf bifurcation taking global awareness rate $\omega$ as the bifurcating parameter. Here, $\lambda = 0.001$ and the rest of the parameters' values are same as Figure 5.

Figure 7 shows the impact of local aware awareness $\eta$. The size of the epidemic decreases accordingly as the rate of awareness increases.

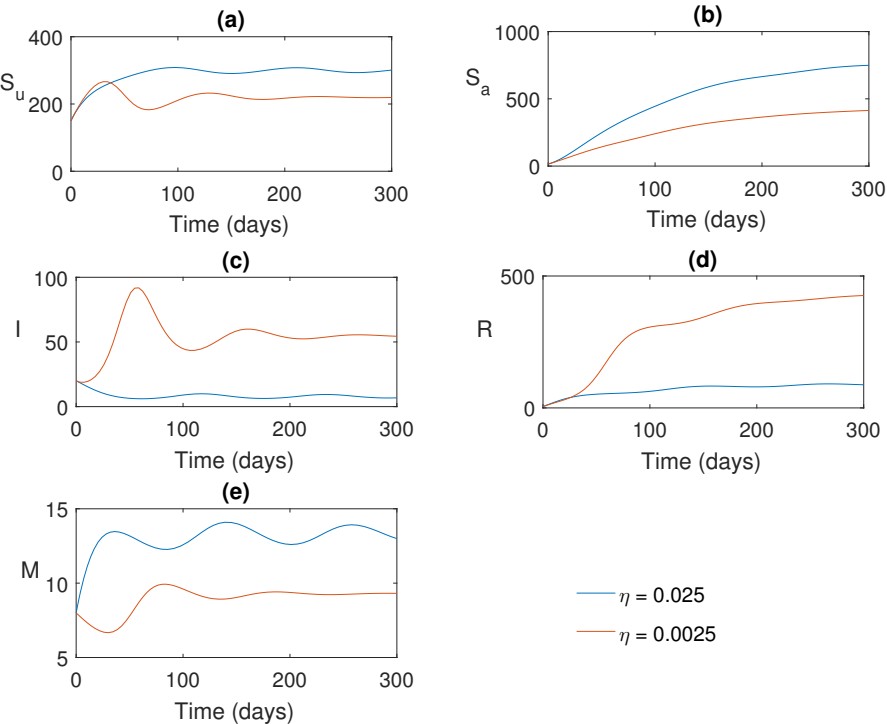

**Figure 7.** (**a**–**e**): Solution trajectories for two different values of the local awareness rate $\eta$.

*Numerical Solution of the Optimal Control Problem*

We used MATLAB to perform the numerical simulations of the optimality system (i.e., system (16) together with system (19) and (24).

The state equations are solved iteratively using an initial guess for the control functions over a desired time interval, employing the fourth-order Runge–Kutta scheme. The adjoint equations are then solved backward in time using the current iteration solutions of the state equations. The control functions are updated using a convex combination of the preceding control functions, and the values from the characterization until the change between the values of unknowns at the earlier iteration and the current iteration is negligible [33].

The numerical simulations of the optimal control problem are shown in Figures 8 and 9. Figure 8a–e illustrate a comparison between the system with and without optimal control. It is evident that optimal control plays a vital role in monitoring the system. The corresponding optimal profiles of the control variables are plotted in Figure 9a,b. The optimal profiles indicate that a high-density awareness campaign is required initially, and treatment is necessary from a later time (after three weeks of disease outbreak).

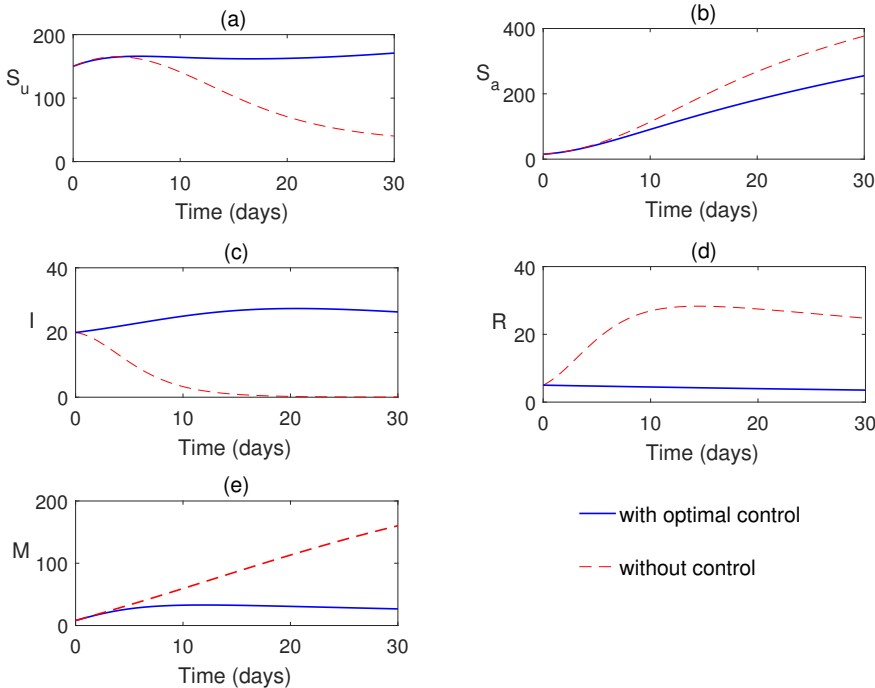

**Figure 8.** (**a**–**e**): Numerical solution of the system (16) with and without control.

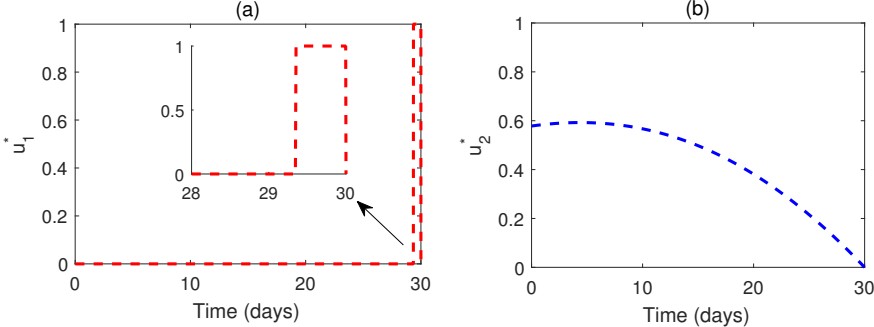

**Figure 9.** (**a**,**b**): Optimal profiles of the optimal controls $u_1^*$ and $u_2^*$.

## 6. Discussion and Conclusions

The global outbreak of infectious diseases has brought about the urgent need for effective control strategies that can reduce their spread while minimizing the economic and social costs associated with control measures. In this study, a mathematical model has been proposed for the prevalence of an infectious disease, taking into account the influence of awareness programs, treatments, and optimal control. This model can help policymakers and public health officials develop effective and cost-efficient control strategies for infectious diseases by incorporating public awareness into the model and using optimization techniques to produce cost-effective control strategies.

Specifically, the proposed awareness-based model is functional and can capture the dynamics of malaria with awareness-based interventions, while the control-induced model can minimize the cost of malaria management. The dynamics of disease propagation have been studied using the proposed mathematical model both analytically and numerically. The next-generation matrix has been used to derive the basic reproduction number $\mathcal{R}_0$. An equilibria assessment shows two equilibria of the proposed model: disease free and endemic. The disease-free equilibrium is stable for $\mathcal{R}_0 < 1$, and the endemic equilibrium exists for $\mathcal{R}_0 > 1$ when the disease-free equilibrium becomes unstable. The endemic equilibrium, existing when $\mathcal{R}_0 > 1$, is asymptotically stable for a lower infection rate. When the infection rate $\lambda$ crosses its critical value $\lambda^* = E^*$ shows Hopf bifurcation.

To further improve the effectiveness and efficiency of control strategies, optimal control theory has been applied to awareness-induced interventions for the cost-effective administration of malaria. The proposed optimal system has been analytically solved using the Pontryagin minimum principle and numerically solved using a specific scheme, which is explained in detail in the study. Optimal profiles of the control variables have been plotted, providing insight into the effect of controls on malaria disease development and the cost sustained in their implementation numerically.

The optimality system consists of six ordinary differential equations (ODEs) from the state and adjoint equations. The optimal solution has been established to be essential and effective in infectious disease control, indicating the potential of this approach for reducing the spread of contagious diseases and minimizing the economic and social costs associated with control measures. Overall, the proposed mathematical model and optimal control strategy can be valuable tools for policymakers and public health officials to develop effective and cost-efficient control strategies for infectious diseases.

To conclude, this article emphasizes the importance of awareness campaigns in controlling the spread of infectious diseases. By educating the public, promoting healthy behaviors, and dispelling misinformation, an effective awareness campaign can help reduce the transmission rate of infectious diseases. Furthermore, the article highlights the role of the World Health Organization in providing resources, guidelines, and tools for the prevention, diagnosis, and treatment of infectious diseases.

The optimal control approach to infectious diseases with awareness-based controls is a promising strategy that can help reduce the spread of contagious diseases while minimizing the economic and social costs associated with implementing control measures. By incorporating public awareness into the model and using optimization techniques to develop cost-effective control strategies, policymakers and public health officials can develop effective and cost-efficient control strategies for infectious diseases. The obtained results from the control-induced model using the maximum principle can be helpful for policymakers in proposing suitable control strategies against infectious diseases.

In a nutshell, this study suggests that awareness campaigns are crucial in controlling infectious diseases, and optimal control theory, combined with media consciousness, is a necessary strategy for infectious disease control. By using this approach, policymakers can develop effective and cost-efficient control strategies that can minimize the economic and social costs associated with controlling infectious diseases.

This study can be extended by including the latent class in the proposed model. A direction and stability analysis of a Hopf bifurcating periodic solution is an important

property of a dynamical system. This can be studied using the proposed model and normal form theory. Considering a time delay due to the latent period or a time delay due to the time required before arranging awareness campaigns, the model can also be extended. We left these ideas as future directions of the present work.

**Author Contributions:** Conceptualization, F.A.B.; methodology, F.A.B., and K.H.; software, F.A.B. and B.R. (Biru Rajak); validation, F.A.B. and K.H.; formal analysis, F.A.B. and K.H.; investigation, F.A.B.; resources, F.A.B. and K.H.; data curation, F.A.B. and K.H.; writing—original draft preparation, B.R. (Bootan Rahman), B.R. (Biru Rajak), and F.A.B.; writing—review and editing, F.A.B. and K.H.; visualization, F.A.B. and K.H.; supervision, F.A.B. All authors have read and agreed to the published version of the manuscript.

**Funding:** This research received no external funding.

**Institutional Review Board Statement:** Not applicable.

**Informed Consent Statement:** Not applicable.

**Data Availability Statement:** Not applicable.

**Acknowledgments:** We are grateful to the reviewers for their comments and suggestions which enrich the paper.

**Conflicts of Interest:** Authors declare that there are no conflict of interest.

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
