# Peer review of "Hopf Bifurcation Analysis and Optimal Control of an Infectious Disease with Awareness Campaign and Treatment"

_axioms, doi:10.3390/axioms12060608_

Round 1

Reviewer 2 Report

The major revision is needed. The most important observation is that the mathematical model (3) is not in perfect accordance with the scheme presented in Figure 1 and must be reformulated.

In the study of the model presented in the paper some corrections are suggested in the attached file.

Concerning the control, one can observe that the cost function was correctly minimized, but the control has negative effects on the population (the number of infected people is larger, the number of recovered people is smaller, the number of unaware susceptible persons is larger.

This aspect must be commented.

The English is very good, however some small corrections are needed.

For example, in the following phrase from abstract I consider that "can maintain system stability" must be replaced by "can maintain the stability of the system" 

"By applying the minimum principle to the optimal control problem, we obtain analytical and numerical results that show how the infection rate of the disease affects the stability of the system and how awareness campaigns and treatment can maintain system stability."

Reviewer 3 Report

In this work, the authors intended to investigate Hopf bifurcation and optimal control of an infectious disease with an awareness campaign and treatment. The equilibrium points are derived. Some numerical results are obtained. However, I have the following comments:

1)      The work is not well presented; Figure 1 is given in the introduction part; No reference for Figure 1 in the text.

2)      The existence and uniqueness of the positive solution for system (1) should be proved.

3)      Where is theorem 3.3 as claimed at the end of Section 3?

4)      In general, the stability analysis and Hopf bifurcation analysis are very weak and incomplete.

5)      Where are the stability analysis of the Hopf bifurcation? i.e. Determine the cases where the Hopf bifurcation is supercritical or subcritical.

6)      In the numerical section:

i)                    Where are the limit cycles arising from the Hopf bifurcation?

ii)                   Where are the system’s complex attractors?

iii)                 The Hopf bifurcation analysis is carried out due to the critical parameter theta; however, the numerical results are provided due to the critical parameter lambda, which is not defined in the theoretical analysis!

Round 2

Reviewer 1 Report

The author addressed some issues in the revised manuscript and I didn't have any major concerns about the improved version, but the authors should still double-check the whole manuscript carefully for some minor typos and/or editing errors. 

Author Response

We thank the reviewer again for the comments and suggestions.

We have rechecked the whole manuscript carefully and corrected the typos and grammatical errors. 

Best regards,

Sincerely,

Fahad

fahadbasir@gmail.com

Reviewer 2 Report

The authors correctly answered to all  observations and made the necessary changes/completions in the text.

Author Response

Thank you for your comments and suggestions. 

Kind regards,

Fahad

fahadbasir@gmail.com

Reviewer 3 Report

In this round, the authors have improved their work according to most of my previous comments. This work can be accepted after the authors fix some minor issues such as:

1) The description of Figure 1 and the citation should be given in Section 2.

2) Theorem ?? in page (9) after equation (9).

3) There is duplication in using the symbol i; on page (9): i = 1, 2, .., 5 and at the same time i is used as a notation of the complex number (-1)^0.5

Author Response

Thank you for your suggestions and comments. We have revised the manuscript accordingly. Point by point response is given below:

Comment 1) The description of Figure 1 and the citation should be given in Section 2.

Response: The description of Figure 1 is now given in section 2.

Comment 2) Theorem ?? in page (9) after equation (9).

Response: The issue is corrected.

3) There is duplication in using the symbol i; on page (9): i = 1, 2, .., 5 and at the same time i is used as a notation of the complex number (-1)^0.5

Response: Suggestion accepted. We use j in place of i.